# Synaptic Sampling: A Bayesian Approach to Neural Network Plasticity and Rewiring

**David Kappel**[1]    **Stefan Habenschuss**[1]    **Robert Legenstein**    **Wolfgang Maass**

Institute for Theoretical Computer Science
Graz University of Technology
A-8010 Graz, Austria
`[kappel, habenschuss, legi, maass]@igi.tugraz.at`

## Abstract

We reexamine in this article the conceptual and mathematical framework for understanding the organization of plasticity in spiking neural networks. We propose that inherent stochasticity enables synaptic plasticity to carry out probabilistic inference by sampling from a posterior distribution of synaptic parameters. This view provides a viable alternative to existing models that propose convergence of synaptic weights to maximum likelihood parameters. It explains how priors on weight distributions and connection probabilities can be merged optimally with learned experience. In simulations we show that our model for synaptic plasticity allows spiking neural networks to compensate continuously for unforeseen disturbances. Furthermore it provides a normative mathematical framework to better understand the permanent variability and rewiring observed in brain networks.

## 1   Introduction

In the 19th century, Helmholtz proposed that perception could be understood as unconscious inference [1]. This insight has recently (re)gained considerable attention in models of Bayesian inference in neural networks [2]. The hallmark of this theory is the assumption that the activity $z$ of neuronal networks can be viewed as an internal model for hidden variables in the outside world that give rise to sensory experiences $x$. This hidden state $z$ is usually assumed to be represented by the activity of neurons in the network. A network $\mathcal{N}$ of stochastically firing neurons is modeled in this framework by a probability distribution $p_{\mathcal{N}}(\mathbf{x}, \mathbf{z}|\boldsymbol{\theta})$ that describes the probabilistic relationships between a set of $N$ inputs $\mathbf{x} = (\boldsymbol{x}^1, \dots, \boldsymbol{x}^N)$ and corresponding network responses $\mathbf{z} = (\boldsymbol{z}^1, \dots, \boldsymbol{z}^N)$, where $\boldsymbol{\theta}$ denotes the vector of network parameters that shape this distribution, e.g., via synaptic weights and network connectivity. The likelihood $p_{\mathcal{N}}(\mathbf{x}|\boldsymbol{\theta}) = \sum_{\mathbf{z}} p_{\mathcal{N}}(\mathbf{x}, \mathbf{z}|\boldsymbol{\theta})$ of the actually occurring inputs $\mathbf{x}$ under the resulting internal model can then be viewed as a measure for the agreement between this internal model (which carries out "predictive coding" [3]) and its environment (which generates $\mathbf{x}$).

The goal of network learning is usually described in this probabilistic generative framework as finding parameter values $\boldsymbol{\theta}^*$ that maximize this agreement, or equivalently the likelihood of the inputs $\mathbf{x}$ (maximum likelihood learning): $\boldsymbol{\theta}^* = \arg\max_{\boldsymbol{\theta}} p_{\mathcal{N}}(\mathbf{x}|\boldsymbol{\theta})$. Locally optimal estimates of $\boldsymbol{\theta}^*$ can be determined by gradient ascent on the data likelihood $p_{\mathcal{N}}(\mathbf{x}|\boldsymbol{\theta})$, which led to many previous models of network plasticity [4, 5, 6]. While these models learn point estimates of locally optimal parameters $\boldsymbol{\theta}^*$, theoretical considerations for artificial neural networks suggest that it is advantageous to learn full posterior distributions $p^*(\boldsymbol{\theta})$ over parameters. This full Bayesian treatment of learning allows to integrate structural parameter priors in a Bayes-optimal way and promises better generalization of the acquired knowledge to new inputs [7, 8]. The problem how such posterior distributions could be learned by brain networks has been highlighted in [2] as an important future challenge in computational neuroscience.

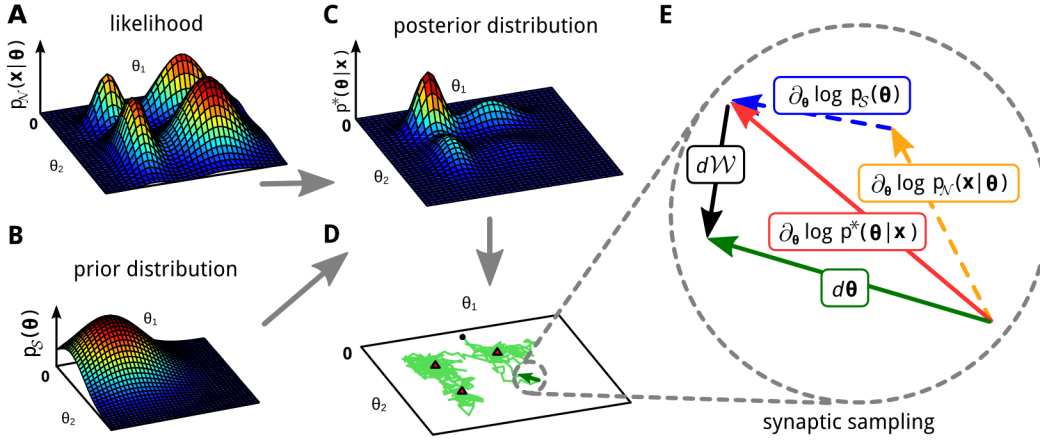

Figure 1: **Illustration of synaptic sampling for two parameters $\boldsymbol{\theta} = \{\theta_1, \theta_2\}$ of a neural network $\mathcal{N}$. A:** 3D plot of an example likelihood function. For a fixed set of inputs $\mathbf{x}$ it assigns a probability density (amplitude on z-axis) to each parameter setting $\boldsymbol{\theta}$. The likelihood function is defined by the underlying neural network $\mathcal{N}$. **B:** Example for a prior that prefers small values for $\boldsymbol{\theta}$. **C:** The posterior that results as product of the prior (B) and the likelihood (A). **D:** A single trajectory of synaptic sampling from the posterior (C), starting at the black dot. The parameter vector $\boldsymbol{\theta}$ fluctuates between different solutions, the visited values cluster near local optima (red triangles). **E:** Cartoon illustrating the dynamic forces (plasticity rule (2)) that enable the network to sample from the posterior distribution $p^*(\boldsymbol{\theta}|\mathbf{x})$ in (D).

Here we introduce a possible solution to this problem. We present a new theoretical framework for analyzing and understanding local plasticity mechanisms of networks of neurons as stochastic processes, that generate specific distributions $p^*(\boldsymbol{\theta})$ of network parameters $\boldsymbol{\theta}$ over which these parameters fluctuate. We call this new theoretical framework *synaptic sampling*. We use it here to analyze and model unsupervised learning and rewiring in spiking neural networks. In Section 3 we show that the synaptic sampling hypothesis also provides a unified framework for structural and synaptic plasticity which both are integrated here into a single learning rule. This model captures salient features of the permanent rewiring and fluctuation of synaptic efficacies observed in the cortex [9, 10]. In computer simulations, we demonstrate another advantage of the synaptic sampling framework: It endows neural circuits with an inherent robustness against perturbations [11].

## 2 Learning a posterior distribution through stochastic synaptic plasticity

In our learning framework we assume that not only a neural network $\mathcal{N}$ as described above, but also a prior $p_{\mathcal{S}}(\boldsymbol{\theta})$ for its parameters $\boldsymbol{\theta} = (\theta_1, \ldots, \theta_M)$ are given. This prior $p_{\mathcal{S}}$ can encode both structural constraints (such as sparse connectivity) and structural rules (e.g., a heavy-tailed distribution of synaptic weights). Then the goal of network learning becomes:

$$\text{learn the posterior distribution:} \quad p^*(\boldsymbol{\theta}|\mathbf{x}) \;=\; \tfrac{1}{\mathcal{Z}} p_{\mathcal{S}}(\boldsymbol{\theta}) \cdot p_{\mathcal{N}}(\mathbf{x}|\boldsymbol{\theta}) \,, \tag{1}$$

with normalizing constant $\mathcal{Z}$. A key insight (see Fig. 1 for an illustration) is that stochastic local plasticity rules for the parameters $\theta_i$ enable a network to achieve the learning goal (1): The distribution of network parameters $\boldsymbol{\theta}$ will converge after a while to the posterior distribution (1) – and produce samples from it – if each network parameter $\theta_i$ obeys the dynamics

$$d\theta_i \;=\; \left( b(\theta_i)\frac{\partial}{\partial\theta_i}\log p_{\mathcal{S}}(\boldsymbol{\theta}) + b(\theta_i)\frac{\partial}{\partial\theta_i}\log p_{\mathcal{N}}(\mathbf{x}|\boldsymbol{\theta}) + T\,b'(\theta_i) \right)\,dt \;+\; \sqrt{2Tb(\theta_i)}\,d\mathcal{W}_i \,, \tag{2}$$

for $i = 1, \ldots, M$ and $b'(\theta_i) = \frac{\partial}{\partial\theta_i}b(\theta_i)$. The stochastic term $d\mathcal{W}_i$ describes infinitesimal stochastic increments and decrements of a Wiener process $\mathcal{W}_i$, where process increments over time $t - s$ are normally distributed with zero mean and variance $t - s$, i.e. $\mathcal{W}_i^t - \mathcal{W}_i^s \sim \text{NORMAL}(0, t - s)$ [12]. The dynamics (2) extend previous models of Bayesian learning via sampling [13, 14] by including a temperature $T > 0$ and a sampling-speed parameter $b(\theta_i) > 0$ that can depend on the current value

of $\theta_i$ without changing the stationary distribution. For example, the sampling speed of a synaptic weight can be slowed down if it reaches very high or very low values.

The temperature parameter $T$ can be used to scale the diffusion term (i.e., the noise). The resulting stationary distribution of $\boldsymbol{\theta}$ is proportional to $p^*(\boldsymbol{\theta})^{\frac{1}{T}}$, so that the dynamics of the stochastic process can be described by the energy landscape $\frac{1}{T}\log p^*(\boldsymbol{\theta})$. For high values of $T$ this energy landscape is flattened, i.e., the main modes of $p^*(\boldsymbol{\theta})$ become less pronounced. For $T = 1$ we arrive at the learning goal (1). For $T \to 0$ the dynamics of $\boldsymbol{\theta}$ approaches a deterministic process and converges to the next local maximum of $p^*(\boldsymbol{\theta})$. Thus the learning process approximates for low values of $T$ maximum a posteriori (MAP) inference [8]. The result is formalized in the following theorem:

**Theorem 1.** *Let $p(\mathbf{x}, \boldsymbol{\theta})$ be a strictly positive, continuous probability distribution over continuous or discrete states $\mathbf{x}$ and continuous parameters $\boldsymbol{\theta} = (\theta_1, \dots, \theta_M)$, twice continuously differentiable with respect to $\boldsymbol{\theta}$. Let $b(\theta)$ be a strictly positive, twice continuously differentiable function. Then the set of stochastic differential equations (2) leaves the distribution $p^*(\boldsymbol{\theta})$ invariant:*

$$p^*(\boldsymbol{\theta}) \equiv \frac{1}{\mathcal{Z}'} p^*(\boldsymbol{\theta} \mid \mathbf{x})^{\frac{1}{T}} , \tag{3}$$

*with $\mathcal{Z}' = \int p^*(\boldsymbol{\theta} \mid \mathbf{x})^{\frac{1}{T}} d\boldsymbol{\theta}$. Furthermore, $p^*(\boldsymbol{\theta})$ is the unique stationary distribution of (2).*

*Proof:* First, note that the first two terms in the drift term of Eq. (2) can be written as

$$b(\theta_i)\frac{\partial}{\partial \theta_i} \log p_{\mathcal{S}}(\boldsymbol{\theta}) + b(\theta_i)\frac{\partial}{\partial \theta_i} \log p_{\mathcal{N}}(\mathbf{x}|\boldsymbol{\theta}) = b(\theta_i)\frac{\partial}{\partial \theta_i} \log p(\theta_i|\mathbf{x}, \boldsymbol{\theta}_{\backslash i}),$$

where $\boldsymbol{\theta}_{\backslash i}$ denotes the vector of parameters excluding parameter $\theta_i$. Hence, the dynamics (2) can be written in terms of an Itô stochastic differential equations with drift $A_i(\boldsymbol{\theta})$ and diffusion $B_i(\boldsymbol{\theta})$:

$$d\theta_i = \left( \underbrace{b(\theta_i)\frac{\partial}{\partial \theta_i} \log p(\theta_i|\mathbf{x}, \boldsymbol{\theta}_{\backslash i}) + T\, b'(\theta_i)}_{\text{drift: } A_i(\boldsymbol{\theta})} \right) dt + \sqrt{\underbrace{2\,T\, b(\theta_i)}_{\text{diffusion: } B_i(\boldsymbol{\theta})}}\; d\mathcal{W}_i . \tag{4}$$

This describes the stochastic dynamics of each parameter over time. For the stationary distribution we are interested in the dynamics of the distribution of parameters. Eq. (4) translate into the following Fokker-Planck equation, that determines the temporal dynamics of the distribution $p_{\mathrm{FP}}(\boldsymbol{\theta}, t)$ over network parameters $\boldsymbol{\theta}$ at time $t$ (see [12]),

$$\frac{d}{dt} p_{\mathrm{FP}}(\boldsymbol{\theta}, t) = \sum_i -\frac{\partial}{\partial \theta_i}\left( A_i(\boldsymbol{\theta})\, p_{\mathrm{FP}}(\boldsymbol{\theta}, t) \right) + \frac{\partial^2}{\partial \theta_i^2}\left( \frac{1}{2} B_i(\boldsymbol{\theta})\, p_{\mathrm{FP}}(\boldsymbol{\theta}, t) \right) . \tag{5}$$

Plugging in the presumed stationary distribution $p^*(\boldsymbol{\theta})$ on the right hand side of Eq. (5), one obtains

$$\frac{d}{dt} p_{\mathrm{FP}}(\boldsymbol{\theta}, t) = \sum_i -\frac{\partial}{\partial \theta_i} \left( A_i(\boldsymbol{\theta})\, p^*(\boldsymbol{\theta}) \right) + \frac{\partial^2}{\partial \theta_i^2} \left( B_i(\boldsymbol{\theta})\, p^*(\boldsymbol{\theta}) \right)$$

$$= \sum_i -\frac{\partial}{\partial \theta_i} \left( b(\theta_i)\, p^*(\boldsymbol{\theta})\, \frac{\partial}{\partial \theta_i} \log p(\theta_i|\mathbf{x}, \boldsymbol{\theta}_{\backslash i}) \right)$$

$$+ \frac{\partial}{\partial \theta_i} \left( T\, b(\theta_i)\, p^*(\boldsymbol{\theta})\, \frac{\partial}{\partial \theta_i} \log p^*(\boldsymbol{\theta}) \right) ,$$

which by inserting for $p^*(\boldsymbol{\theta})$ the assumed stationary distribution (3) becomes

$$\frac{d}{dt} p_{\mathrm{FP}}(\boldsymbol{\theta}, t) = \sum_i -\frac{\partial}{\partial \theta_i} \left( b(\theta_i)\, p^*(\boldsymbol{\theta})\, \frac{\partial}{\partial \theta_i} \log p(\theta_i|\mathbf{x}, \boldsymbol{\theta}_{\backslash i}) \right)$$

$$+ \frac{\partial}{\partial \theta_i} \left( b(\theta_i)\, p^*(\boldsymbol{\theta})\, \frac{\partial}{\partial \theta_i} \left( \log p(\boldsymbol{\theta}_{\backslash i}|\mathbf{x}) + \log p(\theta_i|\mathbf{x}, \boldsymbol{\theta}_{\backslash i}) \right) \right) = \sum_i 0 = 0 .$$

This proves that $p^*(\boldsymbol{\theta})$ is a stationary distribution of the parameter sampling dynamics (4). Under the assumption that $b(\theta_i)$ is strictly positive, this stationary distribution is also unique. If the matrix of diffusion coefficients is invertible, and the potential conditions are satisfied (see Section 3.7.2 in [12] for details), the stationary distribution can be obtained (uniquely) by simple integration. Since the matrix of diffusion coefficients $\boldsymbol{B}$ is diagonal in our model ($\boldsymbol{B} = \mathrm{diag}(B_i(\boldsymbol{\theta}), \dots, B_M(\boldsymbol{\theta}))$), $\boldsymbol{B}$ is trivially invertible since all elements, i.e. all $B_i(\boldsymbol{\theta})$, are positive. Convergence and uniqueness of the stationary distribution follows then for strictly positive $b(\theta_i)$ (see Section 5.3.3 in [12]). □

## 2.1 Online synaptic sampling

For sequences of $N$ inputs $\mathbf{x} = (\boldsymbol{x}^1, \ldots, \boldsymbol{x}^N)$, the weight update rule (2) depends on all inputs, such that synapses have to keep track of the whole set of all network inputs for the exact dynamics (batch learning). In an online scenario, we assume that only the current network input $\boldsymbol{x}^n$ is available.

According to the dynamics (2), synaptic plasticity rules have to compute the log likelihood derivative $\frac{\partial}{\partial \theta_i} \log p_{\mathcal{N}}(\mathbf{x}|\boldsymbol{\theta})$. We assume that every $\tau_x$ time units a different input $\boldsymbol{x}^n$ is presented to the network and that the inputs $\boldsymbol{x}^1, \ldots, \boldsymbol{x}^N$ are visited repeatedly in a fixed regular order. Under the assumption that the input patterns are statistically independent the likelihood $p_{\mathcal{N}}(\mathbf{x}|\boldsymbol{\theta})$ becomes

$$p_{\mathcal{N}}(\mathbf{x}|\boldsymbol{\theta}) = p_{\mathcal{N}}(\boldsymbol{x}^1, \ldots, \boldsymbol{x}^N|\boldsymbol{\theta}) = \prod_{n=1}^{N} p_{\mathcal{N}}(\boldsymbol{x}^n|\boldsymbol{\theta}) \,, \tag{6}$$

i.e., each network input $\boldsymbol{x}^n$ can be explained as being drawn individually from $p_{\mathcal{N}}(\boldsymbol{x}^n|\boldsymbol{\theta})$, independently from other inputs. The derivative of the log likelihood in (2) is then given by $\frac{\partial}{\partial \theta_i} \log p_{\mathcal{N}}(\mathbf{x}|\boldsymbol{\theta}) = \sum_{n=1}^{N} \frac{\partial}{\partial \theta_i} \log p_{\mathcal{N}}(\boldsymbol{x}^n|\boldsymbol{\theta})$. This "batch" dynamics does not map readily onto a network implementation because the weight update requires at any time knowledge of all inputs $\boldsymbol{x}^1, \ldots, \boldsymbol{x}^N$. We provide here an online approximation for small sampling speeds. To obtain an online learning rule, we consider the parameter dynamics

$$d\theta_i = \left( b(\theta_i)\frac{\partial}{\partial \theta_i} \log p_{\mathcal{S}}(\boldsymbol{\theta}) + Nb(\theta_i)\frac{\partial}{\partial \theta_i} \log p_{\mathcal{N}}(\boldsymbol{x}^n|\boldsymbol{\theta}) + Tb'(\theta_i) \right) dt \,+\, \sqrt{2Tb(\theta_i)} \, d\mathcal{W}_i. \tag{7}$$

As in the batch learning setting, we assume that each input $\boldsymbol{x}^n$ is presented for a time interval of $\tau_x$. Although convergence to the correct posterior distribution cannot be guaranteed theoretically for this online rule, we show that it is a reasonable approximation to the batch-rule. Integrating the parameter changes (7) over one full presentation of the data $\mathbf{x}$, i.e., starting from $t = 0$ with some initial parameter values $\boldsymbol{\theta}^0$ up to time $t = N\tau_x$, we obtain for slow sampling speeds ($N\tau_x b(\theta_i) \ll 1$)

$$\theta_i^{N\tau_x} - \theta_i^0 \approx N\tau_x \left( b(\theta_i^0)\frac{\partial}{\partial \theta_i} \log p_{\mathcal{S}}(\boldsymbol{\theta}^0) + b(\theta_i^0)\sum_{n=1}^{N} \frac{\partial}{\partial \theta_i} \log p_{\mathcal{N}}(\boldsymbol{x}^n|\boldsymbol{\theta}^0) + T\,b'(\theta_i^0) \right)$$
$$+ \sqrt{2Tb(\theta_i^0)}\,(\mathcal{W}_i^{N\tau_x} - \mathcal{W}_i^0). \tag{8}$$

This is also what one obtains when integrating the batch rule (2) for $N\tau_x$ time units (for slow $b(\theta_i)$). Hence, for slow enough $b(\theta_i)$, (7) is a good approximation of optimal weight sampling.

In the presence of hidden variables $\mathbf{z}$, maximum likelihood learning cannot be applied directly, since the state of the hidden variables is not known from the observed data. The expectation maximization algorithm [8] can be used to overcome this problem. We adopt this approach here. In the online setting, when pattern $\boldsymbol{x}^n$ is applied to the network, it responds with network state $\boldsymbol{z}^n$ according to $p_{\mathcal{N}}(\boldsymbol{z}^n \,|\, \boldsymbol{x}^n, \boldsymbol{\theta})$, where the current network parameters are used in this inference process. The parameters are updated in parallel according to the dynamics (8) for the given values of $\boldsymbol{x}^n$ and $\boldsymbol{z}^n$.

## 3 Synaptic sampling for network rewiring

In this section we present a simple model to describe permanent network rewiring using the dynamics (2). Experimental studies have provided a wealth of information about the stochastic rewiring in the brain (see e.g. [9, 10]). They demonstrate that the volume of a substantial fraction of dendritic spines varies continuously over time, and that all the time new spines and synaptic connections are formed and existing ones are eliminated. We show that these experimental data on spine motility can be understood as special cases of synaptic sampling. To arrive at a concrete model we use the following assumption about dynamic network rewiring:

1. In accordance with experimental studies [10], we require that spine sizes have a multiplicative dynamics, i.e., that the amount of change within some given time window is proportional to the current size of the spine.

2. We assume here for simplicity that there is a single parameter $\theta_i$ for each potential synaptic connection $i$.

The second requirement can be met by encoding the state of the synapse in an abstract form, that represents synaptic connectivity and synaptic efficacy in a single parameter $\theta_i$. We define that negative values of $\theta_i$ represent a current disconnection and positive values represent a functional synaptic connection (we focus on excitatory connections). The distance of the current value of $\theta_i$ from zero indicates how likely it is that the synapse will soon reconnect (for negative values) or withdraw (for positive values). In addition the synaptic parameter $\theta_i$ encodes for positive values the synaptic efficacy $w_i$, i.e., the resulting EPSP amplitudes, by a simple mapping $w_i = f(\theta_i)$.

The first assumption which requires multiplicative synaptic dynamics supports an exponential function $f$ in our model, in accordance with previous models of spine motility [10]. Thus, we assume in the following that the efficacy $w_i$ of synapse $i$ is given by

$$w_i = \exp(\theta_i - \theta_0). \tag{9}$$

Note that for a large enough offset $\theta_0$, negative parameter values $\theta_i$ (which model a non-functional synaptic connection) are automatically mapped onto a tiny region close to zero in the $w$-space, so that retracted spines have essentially zero synaptic efficacy. In addition we use a Gaussian prior $p_\mathcal{S}(\theta_i) = \mathrm{NORMAL}(\theta_i \,|\, \mu, \sigma)$, with mean $\mu$ and variance $\sigma^2$ over synaptic parameters. In the simulations we used $\mu = 0.5$, $\sigma = 1$ and $\theta_0 = 3$. A prior of this form allows to include a simple regularization mechanism in the learning scheme, which prefers sparse solutions (i.e. solutions with small parameters) [8]. Together with the exponential mapping (9) this prior induces a heavy-tailed prior distribution over synaptic weights $w_i$. The network therefore learns solutions where only the most relevant synapses are much larger than zero.

The general rule for online synaptic sampling (7) for the exponential mapping $w_i = \exp(\theta_i - \theta_0)$ and the Gaussian prior becomes (for constant small learning rate $b \ll 1$ and unit temperature $T = 1$)

$$d\theta_i = b\left(\frac{1}{\sigma^2}\left(\mu - \theta_i\right) + Nw_i \frac{\partial}{\partial w_i} \log p_\mathcal{N}(\boldsymbol{x}^n|\boldsymbol{w})\right)dt + \sqrt{2b}\, d\mathcal{W}_i . \tag{10}$$

In Eq. (10) the multiplicative synaptic dynamics becomes explicit. The gradient $\frac{\partial}{\partial w_i} \log p_\mathcal{N}(\boldsymbol{x}^n|\boldsymbol{w})$, i.e., the activity-dependent contribution to synaptic plasticity, is weighted by $w_i$. Hence, for negative values of $\theta_i$ (non-functional synaptic connection), the activities of the pre- and post-synaptic neurons have negligible impact on the dynamics of the synapse. Assuming a large enough $\theta_0$, retracted synapses therefore evolve solely according to the prior $p_\mathcal{S}(\boldsymbol{\theta})$ and the random fluctuations $d\mathcal{W}_i$. For large values of $\theta_i$ the opposite is the case. The influence of the prior $\frac{\partial}{\partial \theta_i} \log p_\mathcal{S}(\boldsymbol{\theta})$ and the Wiener process $d\mathcal{W}_i$ become negligible, and the dynamics is dominated by the activity-dependent likelihood term.

If the activity-dependent second term in Eq. (10) (that tries to maximize the likelihood) is small (e.g., because $\theta_i$ is small or parameters are near a mode of the likelihood) then Eq. (10) implements an Ornstein-Uhlenbeck process. This prediction of our model is consistent with a previous analysis which showed that an Ornstein-Uhlenbeck process is a viable model for synaptic spine motility [10].

## 3.1 Spiking network model

Through the use of parameters $\boldsymbol{\theta}$ which determine both synaptic connectivity and synaptic weights, the synaptic sampling framework provides a unified model for structural and synaptic plasticity. Eq. (10) describes the stochastic dynamics of the synaptic parameters $\theta_i$. In this section we analyze the resulting rewiring dynamics and structural plasticity by applying the synaptic sampling framework to networks of spiking neurons. Here, we used winner-take-all (WTA) networks to learn a simple sensory integration task and show that learning with synaptic sampling in such networks is inherently robust to perturbations.

For the WTA we adapted the model described in detail in [15]. Briefly, the WTA neurons were modeled as stochastic spike response neurons with a firing rate that depends exponentially on the membrane voltage [16, 17]. The membrane potential $u_k(t)$ of neuron $k$ at time $t$ is given by

$$u_k(t) = \sum_i w_{ki}\, x_i(t) + \beta_k(t) , \tag{11}$$

where $x_i(t)$ denotes the (unweighted) input from input neuron $i$, $w_{ki}$ denotes the efficacy of the synapse from input neuron $i$, and $\beta_k(t)$ denotes a homeostatic adaptation current (see below). The

input $x_i(t)$ models the (additive) excitatory postsynaptic current from neuron $i$. In our simulations we used a double-exponential kernel with time constants $\tau_m = 20\text{ms}$ and $\tau_s = 2\text{ms}$ [18]. The instantaneous firing rate $\rho_k(t)$ of network neuron $k$ depends exponentially on the membrane potential and is subject to divisive lateral inhibition $I_{\text{lat}}(t)$ (described below): $\rho_k(t) = \frac{\rho_{\text{net}}}{I_{\text{lat}}(t)} \exp(u_k(t))$, where $\rho_{\text{net}} = 100\text{Hz}$ scales the firing rate of neurons [16]. Spike trains were then drawn from independent Poisson processes with instantaneous rate $\rho_k(t)$ for each neuron. Divisive inhibition [19] between the $K$ neurons in the WTA network was implemented in an idealized form [6], $I_{\text{lat}}(t) = \sum_{l=1}^{K} \exp(u_l(t))$. In addition, each output spike caused a slow depressing current, giving rise to the adaptation current $\beta_k(t)$. This implements a slow homeostatic mechanism that regulates the output rate of individual neurons (see [20] for details).

The WTA network defined above implicitly defines a generative model [21]. Inputs $\boldsymbol{x}^n$ are assumed to be generated in dependence on the value of a hidden multinomial random variable $h^n$ that can take on $K$ possible values $1, \ldots, K$. Each neuron $k$ in the WTA circuit corresponds to one value $k$ of this hidden variable. One obtains the probability of an input vector for a given hidden cause as $p_{\mathcal{N}}(\boldsymbol{x}^n | h^n = k, \boldsymbol{w}) = \prod_i \text{POISSON}(x_i^n | \alpha e^{w_{ki}})$, with a scaling parameter $\alpha > 0$. In other words, the synaptic weight $w_{ki}$ encodes (in log-space) the firing rate of input neuron $i$, given that the hidden cause is $k$. The network implements inference in this generative model, i.e., for a given input $\boldsymbol{x}^n$, the firing rate of network neuron $z_k$ is proportional to the posterior probability $p(h^n = k | \boldsymbol{x}^n, \boldsymbol{w})$ of the corresponding hidden cause. Online maximum likelihood learning is realized through the synaptic update rule (see [21]), which realizes here the second term of Eq. (10)

$$\frac{\partial}{\partial w_{ki}} \log p_{\mathcal{N}}(\boldsymbol{x}^n \,|\, \boldsymbol{w}) \approx S_k(t) \, (x_i(t) - \alpha \, e^{w_{ki}}) \;, \tag{12}$$

where $S_k(t)$ denotes the spike train of the $k$th neuron and $x_i(t)$ denotes the weight-normalized value of the sum of EPSPs from presynaptic neuron $i$ at time $t$ in response to pattern $\boldsymbol{x}^n$.

## 3.2 Simulation results

Here, we consider a network that allows us to study the self-organization of connections between hidden neurons. Additional details to this experiment and further analyses of the synaptic sampling model can be found in [22].

The architecture of the network is illustrated in Fig. 2A. It consists of eight WTA circuits with arbitrary excitatory synaptic connections between neurons within the same or different ones of these WTA circuits. Two populations of "auditory" and "visual" input neurons $\mathbf{x}_A$ and $\mathbf{x}_V$ project onto corresponding populations $\mathbf{z}_A$ and $\mathbf{z}_V$ of hidden neurons (each consisting of four WTA circuits with $K = 10$ neurons, see lower panel of Fig. 2A). The hidden neuron populations receive exclusively auditory ($\mathbf{z}_A$, 770 neurons) or visual inputs ($\mathbf{z}_V$, 784 neurons) and in addition, arbitrary lateral excitatory connections between all hidden neurons are allowed. This network models multi-modal sensory integration and association in a simplified manner [15].

Biological neural networks are astonishingly robust against perturbations and lesions [11]. To investigate the inherent compensation capability of synaptic sampling we applied two lesions to the network within a learning session of 8 hours (of equivalent biological time). The network was trained by repeatedly drawing random instances of spoken and written digits of the same type (digit *1* or *2* taken from MNIST and 7 utterances of speaker 1 from TI 46) and simultaneously presenting Poisson spiking representations of these input patterns to the network. Fig. 2A shows example firing rates for one spoken/written input pair. Input spikes were randomly drawn according to these rates. Firing rates of visual input neurons were kept fixed throughout the duration of the auditory stimulus.

In the first lesion we removed all neurons (16 out of 40) that became tuned for digit *2* in the preceding learning. The reconstruction performance of the network was measured through the capability of a linear readout neuron, which received input only from $\mathbf{z}_V$. During these test trials only the auditory stimulus was presented (the remaining 3 utterances of speaker 1 were used as test set) and visual input neurons were clamped to $1Hz$ background noise. The lesion significantly impaired the performance of the network in stimulus reconstruction, but it was able to recover from the lesion after about one hour of continuing network plasticity (see Fig. 2C).

In the second lesion all synaptic connections between hidden neurons that were present after recovery from the first lesion were removed and not allowed to regrow (2936 synapses in total). After

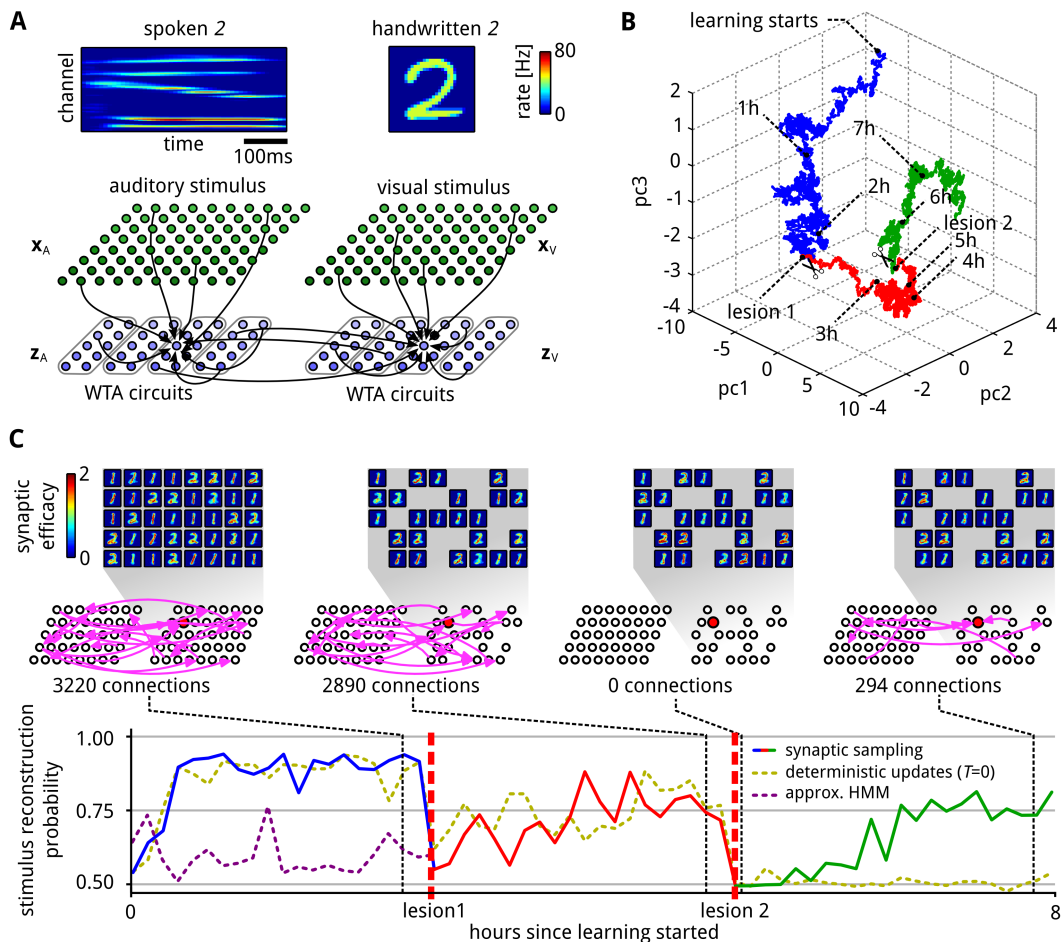

Figure 2: **Inherent compensation for network perturbations. A:** Illustration of the network architecture: A recurrent spiking neural network received simultaneously spoken and handwritten spiking representations of the same digit. **B:** First three PCA components of the temporal evolution a subset of the network parameters $\boldsymbol{\theta}$. After each lesion the network parameters migrate to a new manifold. **C:** The generative reconstruction performance of the "visual" neurons $\mathbf{z}_V$ for the test case when only an auditory stimulus is presented was tracked throughout the whole learning session (colors of learning phases as in (B)). After each lesion the performance strongly degrades, but reliably recovers. Learning with zero temperature (dashed yellow) or with approximate HMM learning [15] (dashed purple) performed significantly worse. Insets at the top show the synaptic weights of neurons in $\mathbf{z}_V$ at 4 time points projected back into the input space. Network diagrams in the middle show ongoing network rewiring for synaptic connections between the hidden neurons. Each arrow indicates a functional connection between two neurons (only 1% randomly drawn subset shown). The neuron whose parameters are tracked in (C) is highlighted in red. Numbers under the network diagrams show the total number of functional connections between hidden neurons at the time point.

about two hours of continuing synaptic sampling 294 new synaptic connections between hidden neurons emerged. These connections made it again possible to infer the auditory stimulus from the activity of the remaining 24 hidden neurons in the population $\mathbf{z}_V$ (in the absence of input from the population $\mathbf{x}_V$). The classification performance was around 75% (see bottom of Fig. 2C).

In Fig. 2B we track the temporal evolution of a subset $\boldsymbol{\theta}'$ of network parameters (35 parameters $\theta_i$ associated with the potential synaptic connections of the neuron marked in red in the middle of Fig. 2C from or to other hidden neurons, excluding those that were removed at lesion 2 and not allowed to regrow). The first three PCA components of this 35-dimensional parameter vector are shown. The vector $\boldsymbol{\theta}'$ fluctuates first within one region of the parameter space while probing

different solutions to the learning problem, e.g., high probability regions of the posterior distribution (blue trace). Each lesions induced a fast switch to a different region (red,green), accompanied by a recovery of the visual stimulus reconstruction performance (see Fig. 2C). The network therefore compensates for perturbations by exploring new parameter spaces.

Without the noise and the prior the same performance could not be reached for this experiment. Fig. 2C shows the result for the approximate HMM learning [15], which is a deterministic learning approach (without a prior). Using this approach the network was able to learn representations of the handwritten and spoken digits. However, these representation and the associations between them were not as distinctive as for synaptic sampling and the classification performance was significantly worse (only first learning phase shown). We also evaluated this experiment with a deterministic version of synaptic sampling ($T = 0$). Here, the stochasticity inherent to the WTA circuit was sufficient to overcome the first lesion. However, the performance was worse in the last learning phase (after removing all active lateral synapses). In this situation, the random exploration of the parameter space that is inherent to synaptic sampling significantly enhanced the speed of the recovery.

## 4   Discussion

We have shown that stochasticity may provide an important function for network plasticity. It enables networks to sample parameters from the posterior distribution that represents attractive combinations of structural constraints and rules (such as sparse connectivity and heavy-tailed distributions of synaptic weights) and a good fit to empirical evidence (e.g., sensory inputs). The resulting rules for synaptic plasticity contain a prior distributions over parameters. Potential functional benefits of priors (on emergent selectivity of neurons) have recently been demonstrated in [23] for a restricted Boltzmann machine.

The mathematical framework that we have presented provides a normative model for evaluating empirically found stochastic dynamics of network parameters, and for relating specific properties of this "noise" to functional aspects of network learning. Some systematic dependencies of changes in synaptic weights (for the same pairing of pre- and postsynaptic activity) on their current values had already been reported in [24, 25, 26]. These can be modeled as the impact of priors in our framework.

Models of learning via sampling from a posterior distribution have been previously studied in machine learning [13, 14] and the underlying theoretical principles are well known in physics (see e.g. Section 5.3 of [27]). The theoretical framework provided in this paper extends these previous models for learning by introducing the temperature parameter $T$ and by allowing to control the sampling speed in dependence of the current parameter setting through $b(\theta_i)$. Furthermore, our model combines for the first time automatic rewiring in neural networks with Bayesian inference via sampling. The functional consequences of these mechanism are further explored in [22].

The postulate that networks should learn posterior distributions of parameters, rather than maximum likelihood values, had been proposed for artificial neural networks [7, 8], since such organization of learning promises better generalization capability to new examples. The open problem of how such posterior distributions could be learned by networks of neurons in the brain, in a way that is consistent with experimental data, has been highlighted in [2] as a key challenge for computational neuroscience. We have presented here a model, whose primary innovation is to view experimentally found trial-to-trial variability and ongoing fluctuations of parameters no longer as a nuisance, but as a functionally important component of the organization of network learning. This model may lead to a better understanding of such noise and seeming imperfections in the brain. It might also provide an important step towards developing algorithms for upcoming new technologies implementing analog spiking hardware, which employ noise and variability as a computational resource [28, 29].

## Acknowledgments

Written under partial support of the European Union project #604102 The Human Brain Project (HBP) and CHIST-ERA ERA-Net (Project FWF #I753-N23, PNEUMA).

We would like to thank Seth Grant, Christopher Harvey, Jason MacLean and Simon Rumpel for helpful comments.

## Footnotes

[1] these authors contributed equally

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
