[Reviews · NeurIPS 2015]

Submitted by Assigned_Reviewer_1

The paper proposes a mechanism for explaining Bayesian inference

and network plasticity in the brain using an algorithm very similar to Stochastic Gradient Langevin Dynamics.

Clarity: The paper is well written. Even though my background is machine learning and not neuroscience, I was able to follow most of the paper.

Originality: The mechanism itself is well studied in the machine learning literature where it is called Stochastic Gradient Langevin Dynamics (SGLD) (see Ref[1] and analysis in Ref[2]). This is also well known in physics where it is usually called the Langevin equation with noisy force (see e.g. section 5.3.'Noisy inexact algorithms' of Ref[3]and references therein). The difference seems to be that the step size (b(\theta) in Eqn. 2) is usually constant in existing work. Also, SGLD has been used only for posterior inference over weights, not for learning the network structure and rewiring.

Significance: Although the proposed mechanism is well known in ML and physics, the paper could be very interesting to the neuroscience community. Both network plasticity and whether the brain does Bayesian inference are important questions, and this seems like a plausible explanation worth investigating more. I would suggest to rewrite the paper so that the mechanism itself is presented as existing work in ML, and focus more on how this is realized in biological neural networks.

Quality: The paper is convincing and shows experimentally how Bayesian inference using the proposed mechanism can help plasticity. Not having much of a neuroscience background, I wasn't able to tell whether this is actually biologically plausible or not, e.g. a) how are priors represented in biological neural networks? b) Also a problem with SGLD is that it is difficult to mix between modes of the posterior distribution. Is this not a problem in biological neural nets?

The paper should cite these (and/or related) references:

[1] Bayesian Learning via Stochastic Gradient Langevin Dynamics, Welling and Teh, ICML 2011

[2] Approximation Analysis of Stochastic Gradient Langevin Dynamics by using Fokker-Planck Equation and Ito Process, Sato and Nakagawa, ICML 2014

[3]The Hybrid Monte Carlo algorithm on parallel computers, A.D. Kennedy, Parallel Computing 1999
Summary: The paper proposes a mechanism for explaining network plasticity and

Bayesian inference in the brain. Although the mechanism itself is well known in ML as Stochastic Gradient Langevin Dynamics, I lean towards accepting the paper because it could be very interesting to the neuroscience community.

Submitted by Assigned_Reviewer_2

Summary The paper describes how fluctuating synaptic efficacy can be interpreted as probabilistic inference on the posterior distribution of synaptic parameters.

Quality The technical details of the paper are beyond my own technical capability to check. While some sampling properties are mathematically proven, the simulation result unfortunately only shows an illustration and not some quantitative measure of (correct) inference.

Clarity On p1, it would probably be helpful to give an intuition as to how P(theta) should be interpreted.

Originality As far as I know, the synaptic sampling models is highly novel, in particular in relation to synaptic motility

Significance The paper combines synaptic plasticity and synaptic sprouting in a single plasticity framework.

One question I had was how the proposed framework relates to the typical quantal nature of synaptic transmission, which would seem to suggest a more discrete stochastic approach.

It also seems interesting to relate the required multiplicative model of synaptic weight updates to known phenomena like synaptic scaling and adaptation.

Typos: l37 neuron -> neurons l51 allows to integrate -> allows us to integrate
Summary: While highly technical, the derivation of synaptic sampling as a Bayesian approach to plasticity and rewiring is - as far as I know - novel and very interesting. Due to space constraints, the illustration feels a bit light in terms of demonstrating the principle.

Author Feedback
Author rebuttal: We thank the reviewers for the detailed feedback. We appreciate that the relevance of our model for computational neuroscience towards a better understanding of the role of noise in the brain was highlighted at several places in the reviews.

We want to specifically address some concerns and potential misunderstandings of the reviewers

To Assigned_Reviewer_2:
"One question I had was how the proposed framework relates to the typical quantal nature of synaptic transmission, which would seem to suggest a more discrete stochastic approach."
We agree that a discrete approach would be interesting. While we believe that such a model could be theoretically tractable, it seems technically more involved and could be considered for future work.

To Assigned_Reviewer_3:
"The basic idea that the stochasticity in spiking and synaptic transmission cam be used for sampling is not particularly novel (Fiser, Langyel, Latham)."
We want to clarify that the proposed model is not primarily based on stochastic synaptic transmission but rather on stochastic synaptic plasticity. This distinguishes the basic idea substantially from previous approaches.

To Assigned_Reviewer_4:
We agree that Stochastic Gradient Langevin Dynamics (SGLD) is well-known in the literature and that this has not been sufficiently been made clear in the first submission. This will be corrected in the final version, together with the proposed citations. We thank the reviewer for pointing out that a novel aspect of our work is the application of SGLD to learning of network structure and rewiring, a fact that will also be explicitly mentioned in the final version.

"how are priors represented in biological neural networks?"
This is of course unknown in general. According to our model, priors (over parameters !) would be represented directly in the plasticity rules, see eq. (2) where the parameter change explicitly depends on the prior distribution p_S(theta).

"Also a problem with SGLD is that it is difficult to mix between modes of the posterior distribution. Is this not a problem in biological neural nets?"
We agree with this statement in general. Note however that the temperature parameter T can be used to facilitate mixing (e.g. by an adaptive scheme) since it can be utilized to flatten the energy landscape. This is described in the paragraph starting at line 110.

To Assigned_Reviewer_7:
We would like to clarify that the manuscript does not only propose a mechanism for network recovery from perturbances. Instead, it provides a framework to understand synaptic plasticity and rewiring as Bayesian inference on the posterior distribution over parameters, which introduces a novel principled approach to these phenomena and has far more implications.

"needs to be improved by (1) reducing presentation clutter (put proofs L127- 171 into appendix and explain the goals and intuitions of the theorems instead..."
We believe that the proof is an essential part of the paper, especially for the NIPS audience which is often also interested in the technical aspects of results. We agree however that the proof can be shortened in favor of the description of intuitions and goals.

"...and (2) adding experiments (repeat the simulations to compute average performance over time; add an experiment explaining *why* sampling increases robustness to disturbances)"
We also performed simulations to study the learning dynamics of synaptic sampling. Our results show that the increased robustness against perturbations stems from the inherent sparseness constraint induced by the prior and permanent exploration due to noise. We evaluated the impact on the learning performance for both of these mechanisms. While this can be noted in the final manuscript, a comprehensive treatment would exceed the space limits and intentions for a conference paper. The goal for this paper was mainly to present the novel concept of synaptic sampling for Bayesian inference over network parameters.